# Si-Based Polarizer and 1-Bit Phase-Controlled Non-Polarizing Beam Splitter-Based Integrated Metasurface for Extended Shortwave Infrared

**DOI:** 10.3390/nano13182592

**Published:** 2023-09-19

**Authors:** Leidong Shi, Lidan Lu, Weiqiang Chen, Guang Chen, Yanlin He, Guanghui Ren, Lianqing Zhu

**Affiliations:** 1School of Instrument Science and Opto-Electronics Engineering, Beijing Information Science and Technology University, Beijing 100192, China; 2021020222@bistu.edu.cn (L.S.); 18363996991@163.com (W.C.); guangchen54@bupt.edu.cn (G.C.); heyanlin@bit.edu.cn (Y.H.); 2Guangzhou Nansha ZiXi Intelligent Sensing Research Institute, Guangzhou 511462, China; 3School of Engineering, RMIT University, Melbourne, VIC 3000, Australia; guanghui.ren@rmit.edu.au

**Keywords:** metasurfaces, polarizer, beam splitter, ESWIR

## Abstract

Metasurfaces, composed of micro-nano-structured planar materials, offer highly tunable control over incident light and find significant applications in imaging, navigation, and sensing. However, highly efficient polarization devices are scarce for the extended shortwave infrared (ESWIR) range (1.7~2.5 μm). This paper proposes and demonstrates a highly efficient all-dielectric diatomic metasurface composed of single-crystalline Si nanocylinders and nanocubes on SiO_2_. This metasurface can serve as a nanoscale linear polarizer for generating polarization-angle-controllable linearly polarized light. At the wavelength of 2172 nm, the maximum transmission efficiency, extinction ratio, and linear polarization degree can reach 93.43%, 45.06 dB, and 0.9973, respectively. Moreover, a nonpolarizing beam splitter (NPBS) was designed and deduced theoretically based on this polarizer, which can achieve a splitting angle of ±13.18° and a phase difference of π. This beam splitter can be equivalently represented as an integration of a linear polarizer with controllable polarization angles and an NPBS with one-bit phase modulation. It is envisaged that through further design optimization, the phase tuning range of the metasurface can be expanded, allowing for the extension of the operational wavelength into the mid-wave infrared range, and the splitting angle is adjustable. Moreover, it can be utilized for integrated polarization detectors and be a potential application for optical digital encoding metasurfaces.

## 1. Introduction

The control capability of electromagnetic waves by natural materials is limited due to the restricted choices in available atoms and their arrangement, as well as the limited range of variation in permittivity and permeability. Metamaterials, consisting of artificial atoms as building blocks, can exhibit arbitrary permittivity and permeability, thereby endowing them with strong modulation capabilities over electromagnetic waves [1,2]. However, due to its three-dimensional complex system, it suffers from challenges such as difficult fabrication, high losses, and propagation phase dependency on material thickness, among others. Optical metasurfaces are derivatives of optical metamaterials, which only require artificial micro-nano structure surfaces to achieve phase mutation at arbitrary interfaces [3,4]. Optical metasurfaces have attracted widespread attention due to their planar structure, ease of fabrication, low loss, and high degree of control [5]. Polarization detection in the ESWIR range holds unique significance. The ESWIR detector can operate at room temperature, avoiding an expensive and sizable refrigerator, and its imaging effect is as good as the reflective imaging of visible light.

Moreover, in addition to its capability to penetrate through the smoke for imaging, ESWIR can detect the details that the mid-wave and long-wave infrared detection lack [6,7]. Polarization detection enables the expansion of information from three dimensions (intensity, spectrum, and spatial) to seven dimensions (intensity, spectrum, spatial, polarization degree, polarization azimuth, polarization ellipticity, and direction of rotation). Therefore, the polarization information of light holds significant importance in scientific domains such as imaging, navigation, and sensing [8,9,10]. However, the integration of detectors with polarization devices remains a challenge in practical applications. Traditional optical polarization devices are often bulky, which is not conducive to the miniaturization of our designed devices and the development of ultra-compact nanophotonics. Metasurfaces, as a promising and efficient alternative tool, can easily address such issues [11]. For instance, metasurfaces based on the surface plasmon resonance effect in subwavelength metal grating structures can be used to design high-extinction-ratio subwavelength metal polarizing filters [12]. However, due to the relatively high optical losses in most metallic materials, this results in generally low transmittance of polarization devices based on metal materials. To tackle this issue, all-dielectric metasurfaces (ADM) relying on the Mie resonance effect have gained significant attention from researchers. ADM not only offers higher transmittance but also provides a convenient platform for integration with semiconductor chips, thanks to the rich diversity of dielectric materials [13]. Indeed, metasurfaces have the capability to apply independent and arbitrary phase distributions to linear, circular, or elliptical polarizations [14]. In previous research, we investigated the application of GaAs-based linear polarizers in the shortwave infrared wavelength range. However, the previously designed polarizers exhibited limited functionality, and GaAs material was not only costlier but also less mature in terms of material growth and etching processes compared to Si. Si has a band gap at 1.1 eV and a cut-off wavelength of around 1110 nm, which is far away from our operating wavelength; therefore, Si has low transmission loss at the ESWIR band. Hence, building upon this groundwork, we propose a novel silicon-based metasurface [15]. Thus, the design of polarizers based on Si with high extinction ratios (ER) that can be utilized in the ESWIR range is a crucial concern in practical applications [16,17].

The NPBS is a fundamental component in high-power laser and optical information processing systems [18,19]. Optical systems are progressing toward the design of multi-branch circuitry and miniaturization [20]. The NPBS serves as a crucial optical component, encompassing functions such as beam division, path selection, and preservation of beam polarization state. Traditional NPBS exist in two forms: multi-layer dielectric films or cubic structures. They are susceptible to laser-induced damage, and their bulky size and orthogonal output direction significantly limit optical layout design and impede system integration [21,22,23]. Optical beam splitting systems based on metasurfaces can effectively address the issues mentioned above. For example, it has been reported that an all-dielectric subwavelength beam splitter, composed of symmetric silicon nanorings, can be designed to manipulate the wavefront phase of the output beams by adjusting the inner and outer radii of the nanorings. This allows for the formation of a phase gradient, and depending on the designed transmittance and angle, it can split any incident polarized light into two beams. By tuning the refractive index of the substrate or adding a silicon film within the substrate, it is possible to alter the initial phase and splitting ratio of the output beams [20]. Additionally, there have been reports on metasurfaces consisting of cross-shaped silicon nano-block arrays placed in a staggered fashion on a silicon dioxide dielectric substrate. These metasurfaces can achieve polarization beam splitting with equal power under the same incident polarized light within the visible wavelength range [24]. However, most beam splitters of this kind, including these examples, may not be suitable for integration due to their fixed splitting ratios, varying beam path lengths, and the use of expensive materials. The recently proposed diatomic metasurface has been widely used in coding, vector holography, polarization control, and optical beam splitting [25,26,27]. However, there is limited research on polarization devices suitable for ESWIR.

This paper demonstrates a nanoscale linear polarizer and an NPBS utilizing a diatomic metasurface of single-crystalline Si, designed explicitly for the ESWIR range. An integrated nano-cube polarization conversion meta-atom (PCM) and a nano-cylinder phase shift meta-atom (PSM) are incorporated in a single unit cell of this metasurface. When properly sized and coordinated, the PCM and PSM can function synergistically as a linear polarization device with high transmittance, ER, and controllable polarization angle. Moreover, the function of NPBS based on supercell is realized by modifying the arrangement of the polarizer’s unit cell, the PCM’s rotation angle, and the PSM’s size. The Jones matrix of metasurface polarizer and NPBS is deduced theoretically. This NPBS effectively imparts a phase delay of π to the transmitted light. In summary, Si-based polarizer and 1-bit phase-controlled NPBS-based integrated metasurface for ESWIR were realized. This metasurface operates in the SWIR wavelength range, and Si-based materials are well-suited for in situ growth with the second-class superlattice InAs/GaSb/AlSb material system. Therefore, this linear polarizer can be used for subsequent integration with polarized infrared detectors based on InAs/GaSb/AlSb superlattices. The NPBS can achieve non-orthogonal splitting, making it potentially valuable for applications in focal plane infrared polarization detection.

## 2. Structure Design and Theoretical Analysis

Figure 1 illustrates the functional and structural schematic of the metasurface. As shown in Figure 1a, a single unit cell of the metasurface polarizer comprises PCM_1_ and PSM_1_ on a SiO_2_ substrate. Based on this, a supercell of NPBS with a π phase difference has been designed. The central region of the supercell, along with its corresponding parameters, is depicted in Figure 1d. The supercell of NPBS is composed of three sets of orthogonal PCM_2_ and PSM_2_ with varying radii. The short axis of PCM_1_ is inclined at an angle α to the horizontal direction, while the short axis of PCM_2_ is inclined at an angle α_2_.

### 2.1. Analysis of the Design Principle of the Polarizer with Arbitrary Polarization Control Angle

Arbitrary optical components can be represented by a transmission matrix, denoted as *M*.
(1)$$M=\left(\begin{array}{cc} J_{XX} & J_{XY}\\ J_{YX} & J_{YY} \end{array}\right)$$

Represent the Jones matrix in terms of amplitude and phase. Thus, the metasurface Jones matrix of symmetrical structures can be expressed as:(2)$$M_{1}=\left(\begin{array}{cc} \left|t_{xx}\right|e^{i\varphi_{xx}} & 0\\ 0 & \left|t_{yy}\right|e^{i\varphi_{yy}} \end{array}\right)$$

By imposing a rotation angle *α* relative to the horizontal direction of the structure, the Jones matrix can be described as:(3)$$M_{2}=\left(\begin{array}{cc} cos\alpha & -sin\alpha \\ sin\alpha & cos\alpha \end{array}\right)\left(\begin{array}{cc} \left|t_{xx}\right|e^{i\varphi_{xx}} & 0\\ 0 & \left|t_{yy}\right|e^{i\varphi_{yy}} \end{array}\right)\left(\begin{array}{cc} cos\alpha & sin\alpha \\ -sin\alpha & cos\alpha \end{array}\right)$$

According to the unique properties of *PSM*_1_ and *PCM*_1_, their Jones matrices can be expressed as $M_{PSM_{1}}$ and $M_{PCM_{1}}$.
(4)$$M_{PSM_{1}}=\left(\begin{array}{cc} \left|t_{xx\_PSM_{1}}\right|e^{i\varphi_{xx\_PSM_{1}}} & 0\\ 0 & \left|t_{xx\_PSM_{1}}\right|e^{i\varphi_{xx\_\_PSM_{1}}} \end{array}\right)$$
(5)$$M_{PCM_{1}}=\left(\begin{array}{cc} \left|t_{xx{\_}_{PCM_{1}}}\right|e^{i\varphi_{xx{\_}_{PCM_{1}}}}\left(cos^{2}\alpha -sin^{2}\alpha \right) & 2\left|t_{xx{\_}_{PCM_{1}}}\right|e^{i\varphi_{xx{\_}_{PCM_{1}}}}cos\alpha sin\alpha \\ 2\left|t_{xx{\_}_{PCM_{1}}}\right|e^{i\varphi_{xx{\_}_{PCM_{1}}}}cos\alpha sin\alpha & \left|t_{xx\_PCM_{1}}\right|e^{i\varphi_{xx\_PCM_{1}}}\left(sin^{2}\alpha -cos^{2}\alpha \right) \end{array}\right)$$

The Jones matrix concentrated in the unit cell is $M_{UC_{1}}$.
(6)$$M_{UC_{1}}=M_{PSM_{1}}+M_{PCM_{1}}=2\left|t_{xx\_\_PSM_{1}}\right|e^{i\varphi_{xx\_PSM_{1}}}\left(\begin{array}{cc} cos^{2}\alpha & sin\alpha cos\alpha \\ sin\alpha cos\alpha & sin^{2}\alpha \end{array}\right)$$

After the incident light $J_{in}=\left(\begin{array}{c} A_{ox}e^{i\varphi_{ox}}\\ A_{oy}e^{i\varphi_{oy}} \end{array}\right)$ with elliptical polarization passes through the metasurface, the Jones matrix of the transmitted light can be expressed as:(7)$$J_{out}=M_{UC_{1}}J_{in}=2\left|t_{xx\_PSM_{1}}\right|e^{i\varphi_{xx\_PSM_{1}}}\left(A_{ox}e^{i\varphi_{ox}}cos\alpha +A_{oy}e^{i\varphi_{oy}}sin\alpha \right)\left(\begin{array}{c} cos\alpha \\ sin\alpha \end{array}\right)$$

Based on the analysis, it can be concluded that the metasurface is composed of $PSM_{1}$ and $PCM_{1}$ can be effectively equivalent to a linear polarizer with an adjustable polarization angle equal to the rotation angle *α* of $PCM_{1}$.

The Jones matrix theoretical derivation of the diatomic metasurface polarizer has been given in previous studies. Ref. [15] shows details.

### 2.2. Analysis of the Design Principle of the NPBS with a Phase Difference of π

By carefully designing the rotation angle of $PCM_{1}$ and the dimensions of $PSM_{1}$, both the matrix representation of $M_{UC1}$ and $\varphi_{xx\_PSM_{1}}$ corresponding to $PSM_{1}$ can be altered. Hence, the phase modulation capability can be realized with this diatomic metasurface. Based on this principle, further implementation of an NPBS function can be achieved.

By applying a rotation angle $\alpha_{2}=\alpha +90^{{}^{\circ}}$ relative to the horizontal direction of $PCM_{1}$, the Jones matrix $M_{PCM1}$ in Equation (5) can be modified to:(8)$$M_{PCM_{2}}=\left(\begin{array}{cc} \left|t_{xx{\_}_{PCM_{1}}}\right|e^{i\varphi_{xx{\_}_{PCM_{1}}}}\left(sin^{2}\alpha -cos^{2}\alpha \right) & -2\left|t_{xx{\_}_{PCM_{1}}}\right|e^{i\varphi_{xx{\_}_{PCM_{1}}}}cos\alpha sin\alpha \\ -2\left|t_{xx{\_}_{PCM_{1}}}\right|e^{i\varphi_{xx{\_}_{PCM_{1}}}}cos\alpha sin\alpha & \left|t_{xx\_PCM_{1}}\right|e^{i\varphi_{xx\_PCM_{1}}}\left(cos^{2}\alpha -sin^{2}\alpha \right) \end{array}\right)$$

If $PSM_{1}$ remains unchanged, according to Equations (4) and (8), the Jones matrix of the newly designed metasurface, denoted as $M_{UC\alpha_{2}}$, can be obtained.
(9)$$M_{UC\alpha_{2}}=2\left|t_{xx_{\_PSM_{1}}}\right|e^{i\varphi_{xx{\_}_{PSM_{1}}}}\left(\begin{array}{cc} sin^{2}\alpha & -sin\alpha cos\alpha \\ -sin\alpha cos\alpha & cos^{2}\alpha \end{array}\right)$$

As inferred from the Jones matrices $M_{UC_{1}}$ and $M_{UC\alpha_{2}}$ in Equations (6) and (9), the two designed metasurface linear polarizers have different polarization angles ($\alpha_{2}=\alpha +90^{{}^{\circ}}$).

If considering the design of a new $PSM_{2}$. By varying the radius of $PSM_{2}$ to provide different phase gradients, it is possible to establish a relationship between the designed $PSM_{1}$ and $PSM_{2}$ in terms of $\left|t_{xx_{-PSM_{2}}}\right|=\left|t_{xx{\_}_{PSM_{1}}}\right|=\left|t_{xx{\_}_{PCM_{1}}}\right|$ and $\varphi_{xx\_PSM_{2}}=\varphi_{xx\_PCM_{1}}+\pi =\varphi_{yy\_PCM_{1}}=\varphi_{xx\_PSM_{1}}+\pi$. Based on Equations (4) and (8), and the conditions mentioned above, the Jones matrix of the newly designed metasurface can be obtained as follows:(10)$$M_{UC_{2}}=M_{PSM_{2}}+M_{PCM_{2}}=2\left|t_{xx\_PSM_{1}}\right|e^{i(\varphi_{xx\_PSM_{1}}+\pi )}\left(\begin{array}{cc} cos^{2}\alpha & sin\alpha cos\alpha \\ sin\alpha cos\alpha & sin^{2}\alpha \end{array}\right)$$

The conclusion can be drawn from Equations (6) and (10). The metasurface formed by the combination of *PCM* with a 90° difference in rotation angle and *PSM* with varying radii exhibits identical polarization modulation functionality while inducing a phase gradient of π.

According to the generalized Snell’s law, it is possible to arbitrarily control the propagation path of refracted and reflected light rays by manipulating the phase gradient of refraction and reflection, due to the devised two types of metasurface structures, distinct phase gradients can be provided. Therefore, it is feasible to integrate the two different metasurface structures within a single supercell to achieve beam splitting of light rays. Therefore, the theoretical refractive angles of NPBS for incident x and y-polarized light can be expressed as follows:(11)$$\theta_{t}=\pm \arcsin\left[\frac{1}{n_{t}}\left(n_{i}\sin\theta_{i}+\frac{\lambda_{0}}{\wedge_{x}}\right)\right]$$
where $\lambda_{0}$ is the wavelength of incident light in free space; $\theta_{i}$ is the angle of incidence; $\wedge_{x}$ is the period of the designed metasurface supercell in the x-direction; $n_{i}$ and $n_{t}$ are the refractive indices of the incident medium and transmitted medium, respectively. Therefore, the final theoretical transmission angle is $\theta_{t}=\pm \arcsin(\frac{\lambda_{0}}{\wedge_{x}}$). The integration of a polarizer and NPBS can be achieved by designing a metasurface based on the aforementioned principles. The schematic diagram of the transmission matrix of metasurface NPBS is shown in Figure 2.

## 3. Numerical Simulation and Optimization

To validate the aforementioned theory and structure, an all-dielectric diatomic metasurface was constructed and evaluated using the Finite-Difference Time-Domain (FDTD) method. Simulation validation was conducted using the FDTD Solutions software 2020 developed by Lumerical Solutions, Canada. A plane wave was utilized as the incident light source, while periodic boundary conditions were employed in both the x and y directions to simulate an extensive array of diatomic metasurfaces. In the z direction, Perfectly Matched Layers (PML) were selected to absorb all incident fields without generating any reflections. For the simulations of the polarizer and NPBS, PML layers with thicknesses of 64 were chosen. The mesh size in the FDTD method significantly influences simulation outcomes. A smaller mesh size enables more precise capture of fine details in optical fields or electromagnetic waves but comes with increased computational complexity and resource demands. Conversely, larger mesh sizes can reduce computational costs but might compromise the accurate representation of high-frequency details and small-scale structures. In simulations, exploring various mesh sizes and comparing their results is essential to determine an appropriate mesh dimension. Additionally, factors such as the nature of the structure, wavelength, and the physical phenomena being simulated should be considered when selecting the suitable mesh size. In the simulation of this metasurface, we employed an ‘auto non-uniform’ mesh type. The mesh accuracy was set to 2, and the mesh refinement option selected was the conformal variant 0. For the time step, a stability factor of 0.99 was utilized. The minimum mesh step size was set to 0.25 nm.

### 3.1. The Metasurface Polarizer

To validate the operation of the polarization-dependent metasurface in the ESWIR range, a planar light source with a working wavelength ranging from 2160 to 2210 nm was selected as the incident light. The incident polarization angle of the light source was systematically varied from 0° to 180° through parameter scanning. The transmittance of the polarizer and the near-field coupling effect are both influenced by the appropriate unit cell period. Therefore, the metasurface’s cell size (P1) was designed as 1500 nm. ${\mathrm{PSM}}_{1}$ and ${\mathrm{PCM}}_{1}$ were placed at opposite corners of the unit cell, with their respective dimensions (r, H, L, W) set to 243 nm, 500 nm, 600 nm, and 378 nm. The polarization modulation effect of the metasurface under different rotation angles was verified by designing α as 45° and 135°. The obtained results are shown in Figure 3.

As shown in Figure 3, the metasurface exhibits distinct transmission responses to light with different incident polarization angles. At α = 45°, the maximum transmittance (Ta) and minimum transmittance (Ti) are observed for incident polarization angles of 45° and 135°, respectively. Conversely, at α = 135°, the opposite trend is observed. These results are consistent with our previous theoretical analysis. The Ta and Ti are defined as the transmittance when the polarization angle of the light source is parallel and perpendicular to the short axis of ${\mathrm{PCM}}_{1}$, respectively. From Figure 3, it can be observed that the best polarization modulation effect occurs at a wavelength of 2172 nm. Based on the design specification of the metasurface cell (the unit cell period is approximately half of the operating wavelength), the operating wavelength can be extended to mid-wave infrared by optimizing the unit cell size. The corresponding transmittance and ER data at this wavelength are presented in Table 1. The ER, obtained by calculating ER = 10 × lg(Ta/Ti), is also provided. Compared to conventional nanopatterned metallic gratings that often achieve transmission rates of just over 70%, the designed silicon-based polarizer attains a maximum transmission exceeding 90%. The low-loss characteristics of Si substrate materials significantly enhance the filtering performance of the polarizer.

Based on the aforementioned data, it can be inferred that $PCM_{1}$ with different rotation angles enable different polarization modulations, achieving optimal performance at a wavelength of 2172 nm. However, this metasurface operates in synergy with ${\mathrm{PSM}}_{1}$. Previous theoretical analysis has revealed that ${\mathrm{PSM}}_{1}$ with varying radii also influences the performance of the metasurface. Considering previous experimental results, we select a fixed wavelength of 2172 nm and investigate the influence of ${\mathrm{PSM}}_{1}$’s radius by varying it. The simulation results are depicted in Figure 4.

According to the figure above, the variation in the radius of ${\mathrm{PSM}}_{1}$ leads to a significant change in the functionality of the polarizer. When α is 45°, the incident polarization angle at which the transmittance reaches its minimum value changes from 135° to 45° as the radius increases. This implies that as the ${\mathrm{PSM}}_{1}$ radius increases, the metasurface undergoes a transition in functionality from a +45° linear polarization state to a −45° linear polarization state. Conversely, when α is 135°, the result is exactly opposite to that at 45°.

To visually illustrate the impact of PSM_1_ radius on the polarization state, Figure 4c,d depict the transmittance and ER of the polarizer. When the PSM_1_ radius is 240 nm, for α values of 45° and 135°, the Ta and ER are determined to be 93.43%, 93.45%, 45.06, and 44.19, respectively. It can be observed from the figure that when the PSM_1_ radius exceeds 257 nm, the ER becomes negative. This is attributed to the fact that the defined Ta is smaller than Ti. It is precisely this change in polarization modulation functionality and the generation of positive and negative ER that offer design insights and possibilities for NPBS. An important point to note is that here, the extinction ratio of this design for metasurfaces can reach up to 45 dB. Compared to our previous metasurface design based on GaAs materials, there is a significant performance improvement [15].

To better understand the polarization state of the light after passing through a polarizer, we fix the polarization angle of the incident light source at 0° and introduce the polarization ellipse and its associated parameters, as shown in Figure 5. The polarization ellipse is a highly effective and intuitive method for describing the polarization state of the transmitted light. The major angle is defined as the angle between the major axis of the polarization ellipse and the horizontal direction. It characterizes the polarization direction of the transmitted polarized light and also reflects the modulation of the polarization angle of the incident light by the linear polarization device. The degree of polarization is a physical quantity that describes the polarization state of a light beam. It is used to quantitatively analyze the polarization and non-polarization components in a light beam. For perfectly linearly polarized light, the Degree of Linear Polarization (DOLP) equals 1. It can be calculated using the formula DOIP = 1 − 2γ/(1 + $\gamma^{2}$), where *γ* represents the ratio of the major axis to the minor axis of the polarization ellipse. The term “phase_diff” refers to the phase difference between the s-polarized component and the p-polarized component. Theoretically, for linearly polarized light, the phase difference is either 0° or 180°.

As shown in Figure 5c,d, the variation of ${\mathrm{PSM}}_{1}$ radius has a significant impact on the polarization ellipse. It is worth noting the gray bar region in Figure 5. When α is 45° and ${\mathrm{PSM}}_{1}$ radius is 240 nm, the DOLP, major angle, and Phase_diff are measured as 0.9973, 45.06°, and 0°, respectively. When α is 135° and ${\mathrm{PSM}}_{1}$ radius is 286 nm, the DOLP, major angle, and Phase_diff are measured as 0.9571, 46.34°, and 0°, respectively. The presented data reveals that the polarization direction of the transmitted light through each of the two metasurfaces closely approaches 45°. This observation is consistent with our previous theoretical derivations, establishing its relevance for designing NPBS.

In summary, the simulated data results presented above demonstrate that by adjusting the rotation angle of the PCM, various polarization modulation effects can be achieved. Additionally, optimizing the radius of the PSM enhances the transmittance, ER, and DOLP of the polarizer.

### 3.2. The Metasurface NPBS

Our theoretical analysis has established that metasurface linear polarizers with different radii exhibit the ability to generate polarized light with the same polarization angle but varying phase differences upon transmission. Building upon this foundation, a new unit cell structure, as depicted in Figure 1d, was designed for the beam splitter. The structural parameters of the PCM remain the same as before, with a periodicity of P2 = 1600 nm. The designed ${\mathrm{PCM}}_{1}$, with α = 45°, and ${\mathrm{PCM}}_{2}$, with α = 135°, correspond to ${\mathrm{PSM}}_{1}$ and ${\mathrm{PSM}}_{2}$ with radii of 240 nm and 286 nm, respectively. Three sets of orthogonal ${\mathrm{PCM}}_{2}$ units, combined with different radii of ${\mathrm{PCM}}_{2}$, form a supercell with a periodicity of $\wedge_{x}$ = 9600 nm in the x-direction. The periodic arrangement of these supercells constitutes the metasurface beam splitter. In terms of functionality, this beam splitter can be regarded as an integrated linear polarizer and an NPBS.

Incident light sources with different polarization directions of 0°, 45°, 90°, and 135°, all at a wavelength of 2172 nm, were used as the input light source. Periodic boundary conditions were applied in the x and y directions, while a PML boundary condition with a layer thickness of 64 was employed in the z-direction. For the calculation of the approximate far-field projection, 300 periods and incident polarization angles of 135° were considered. The simulation results are presented in Figure 6a. Under normal incidence, the splitting angle can be theoretically calculated using the generalized Snell’s law, yielding a theoretical splitting angle of $\theta_{t}=\pm \arcsin[\frac{2172}{9600}$] = $\pm 13.08^{{}^{\circ}}$. The simulated splitting angle achieved approximately ±13.18°. Additionally, Figure 6b–e provide simulated far-field patterns of transmitted intensity for incident polarization angles of 0°, 45°, 90°, and 135°.

From Figure 6b–e, it can be observed that when the incident polarization angle is 0° or 90°, the transmitted light has a relatively lower intensity, approximately half of that at a polarization angle of 45°. When the incident polarization angle is 135°, the transmitted power becomes minimal, even reaching 0. These observations are consistent with the definition of the Malus law regarding the intensity of transmitted light. When the incident polarization angle is 45°, the incident light transmits to two different directions with relatively higher intensity, exhibiting an inclined angle of approximately ±13.18°. This phenomenon is also following Snell’s law. Compared to conventional NPBS capable of producing only orthogonal right-angle outputs, the introduced ultra-thin metasurface NPBS, with the capability to achieve ±13.18° divergence angles, presents a novel beam-splitting approach. This innovation broadens the spectrum of choices available for practical applications.

However, it is worth noting that although the polarization purity of the split beams is relatively high, their actual intensity is lower. This is because, in addition to the desired diffraction order, the transmitted light can be found in many other diffraction orders. To address this issue, a larger number of periods, specifically 5000, were considered when calculating the approximate far-field projection. The simulation results are presented in Figure 6f–i. It can be observed that the light from diffraction orders other than ±13.18° is significantly weakened or even disappears, and the overall intensity of the polarized light after splitting is noticeably enhanced. From the simulated results in Figure 6, it can be observed that when the unit cell period is chosen as 300 and the incident polarization angle is 45°, the maximum intensity of transmitted polarized light is 1.82 × 10^−4^. Conversely, for an incident polarization angle of 135°, the weakest intensity of transmitted polarized light is 1.15 × 10^−5^. This indicates that the designed beam splitter indeed exerts varying modulation effects on incident light with different polarization angles. Additionally, when the incident polarization angle is 0° and 90°, the intensities of incident polarized light are approximately 9.77 × 10^−5^ and 9.66 × 10^−5^, respectively. The intensities at 0° and 90° are roughly half of that at 45°, following the definition of polarized transmitted light intensity following Malus’s Law. On the other hand, it is worth noting that as the number of unit cells increases to 5000, the excessive diffraction orders weaken or even disappear. At this point, with an incident polarization angle of 45°, the strongest transmitted polarized light has an intensity of 5.81 × 10^−3^. Compared to the unit cell period of 300, the transmitted polarized light intensity increases by an order of magnitude. This demonstrates that optimizing the number of unit cells can effectively enhance the intensity of transmitted light and reduce unnecessary diffraction orders. Therefore, increasing the number of supercells in the metasurface can significantly improve the issues associated with the beam splitter.

In conclusion, our designed metasurface NPBS can achieve beam splitting for light with the same polarization direction. However, having too few unit cells in the metasurface NPBS leads to unnecessary diffraction orders and reduces the intensity of transmitted light. Therefore, we optimized the design by significantly increasing the number of unit cells across a large area. This optimization reduces the excess diffraction orders in the metasurface NPBS, thereby greatly enhancing the intensity of the desired transmitted light.

Snell’s Law theoretically explains the phase difference between two polarized light beams, resulting in a simulated splitting angle of 13.18°. Building upon the aforementioned data, the simulation explores the phase difference induced by a metasurface when two polarized light beams are incident upon it. By incorporating field monitors along the transmission path, the phase of the transmitted light is obtained within the supercell. Figure 7 depicts the spatial distribution of the phase of the polarized light, with the X and Y axes representing the supercell’s dimensions. The distinct structures in the left and right halves of the supercell yield a phase difference close to π for the polarized light. Moreover, different sizes of PSM will give different phase gradients to the metasurface, so it is possible to optimize the metasurface parameters by design to achieve NPBS with different splitting angles. Compared with the traditional NPBS with an orthogonal splitting angle, the method proposed in this paper provides a new scheme for the splitting angle adjustment of NPBS.

The experimental results presented above further demonstrate that our designed metasurface functions as an integrated device, equivalent to a polarization-controllable linear polarizer operating in the ESWIR wavelength range and an NPBS with a phase difference of π. In other words, based on the proposed dielectric diatomic metasurface, we can easily achieve 1-bit phase control, where the states “0” and “1” correspond to phase values of “0” and “π”, respectively. This all-dielectric diatomic metasurface, capable of achieving 1-bit phase modulation, can be considered an effective platform for manipulating electromagnetic waves in the form of a digital encoding metasurface. We envision that by further optimizing the diatomic metasurface, its phase tuning range can be expanded, allowing for the realization of diverse metasurface devices with various wavefront shaping functionalities. Such devices hold promising potential for applications in optical digital encoding metasurfaces [28].

Some mature processes have processed the metasurface proposed above. The existing silicon-based optical waveguide technology has reached a considerable level of maturity, rendering it of paramount significance in guiding the fabrication processes of the silicon-based material metasurface that we have devised [29]. Si films can be grown on SiO_2_ substrates by molecular beam epitaxy (MBE) technology. The desired diatomic metasurface pattern is then processed by electron beam exposure (EBL). Finally, the whole structure was fabricated by inductively coupled plasma (ICP) etching technology using HF as the etching gas [30,31].

## 4. Conclusions

In conclusion, this paper presents a single-layer, all-dielectric diatomic metasurface composed of PCM and PSM based on monocrystalline Si operating in the ESWIR wavelength range (1.7~2.5 μm). Through rigorous theoretical analysis and numerical simulations using the FDTD method, the operating wavelength, transmission efficiency, ER, and DOLP of the polarizer are systematically studied about incident polarization angle, PSM radius (220~300 nm), and PCM rotation angle (α = 45° and 135°). The results demonstrate that the designed metasurface can function as a linear polarizer with the polarization angle dependent on the PCM rotation angle. The maximum transmission efficiency of the polarizer reaches 93.43%, with an ER of 45.06 dB and a DOLP of 0.9973. Furthermore, based on this design, an NPBS is proposed, which can achieve an actual splitting angle of ±13.18°. The proposed metasurface can be equivalently integrated as a linear polarizer with customizable polarization angles and an NPBS with an output light having a phase difference of π. The metasurface based on Si material has a relatively mature process to achieve results. It is envisaged that through further design optimization, the phase tuning range of the metasurface can be expanded, allowing for the extension of the operational wavelength into the mid-wave infrared range, and the splitting angle is adjustable. Moreover, this diatomic metasurface, featuring 1-bit phase modulation capability, holds potential as a prospective application for optical digital encoding metasurfaces.

## Figures and Tables

**Figure 1 nanomaterials-13-02592-f001:**
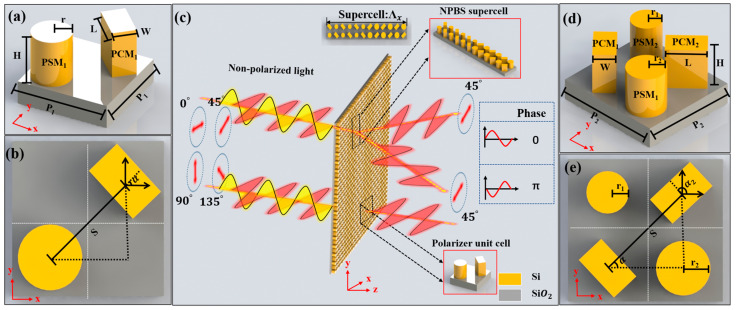
Perspective (**a**) and top view (**b**) of a single unit cell of the linear polarizer, schematic diagram of the functionality and structure of the metasurface (**c**), perspective (**d**) and top view (**e**) of the central portion of the supercell of NPBS.

**Figure 2 nanomaterials-13-02592-f002:**
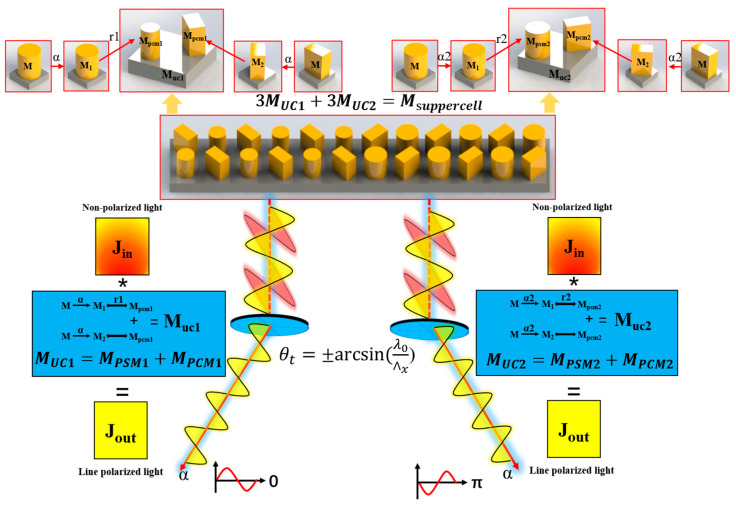
Schematic diagram of the transmission matrix of metasurface NPBS.

**Figure 3 nanomaterials-13-02592-f003:**
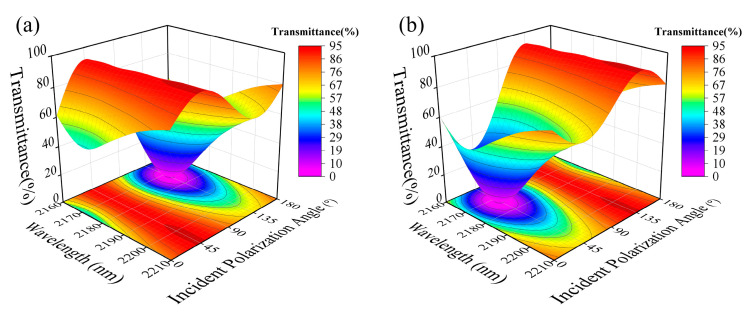
Transmittance as a function of wavelength and incident polarization angle at α = 45° (**a**) and 135° (**b**).

**Figure 4 nanomaterials-13-02592-f004:**
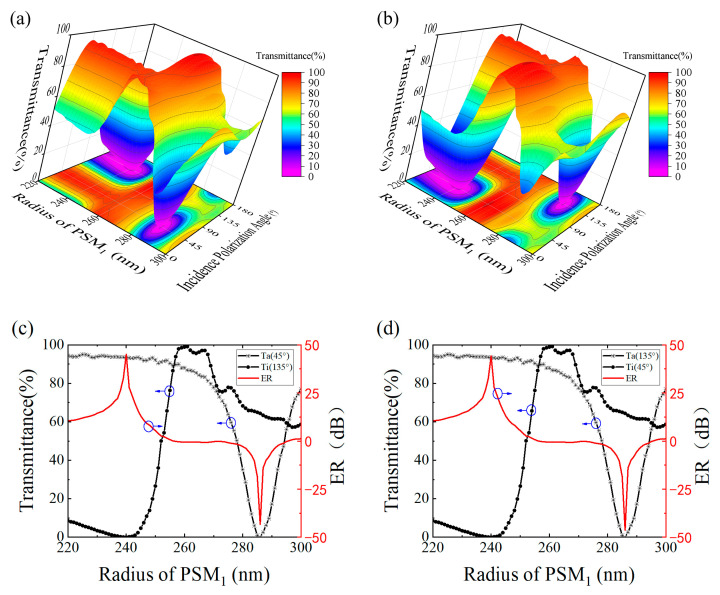
Transmittance as a function of the radius of *PSM*_1_ and incident polarization angle at α = 45° (**a**) and 135° (**b**). Variation of transmittance and ER with *PSM*_1_ radius for α = 45° (**c**) and α = 135° (**d**).

**Figure 5 nanomaterials-13-02592-f005:**
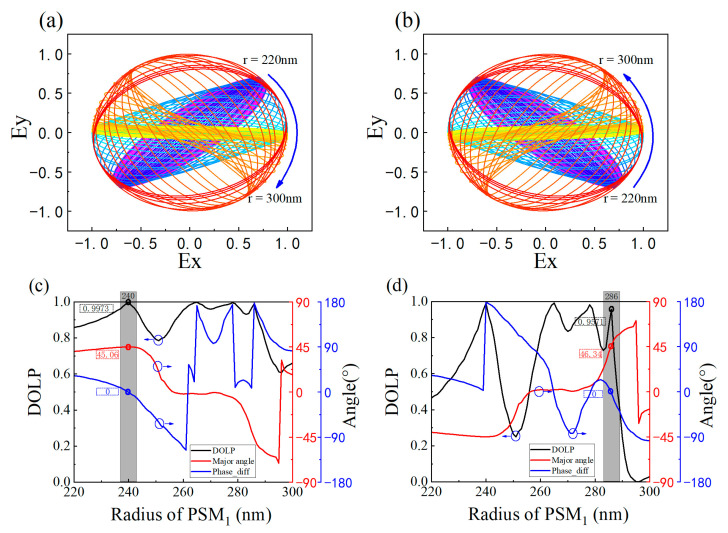
The polarization ellipses of transmitted light for α = 45° (**a**) and α = 135° (**b**). Variation of DOLP, ER, and Phase_diff with PSM_1_ radius for α = 45° (**c**) and α = 135° (**d**).

**Figure 6 nanomaterials-13-02592-f006:**
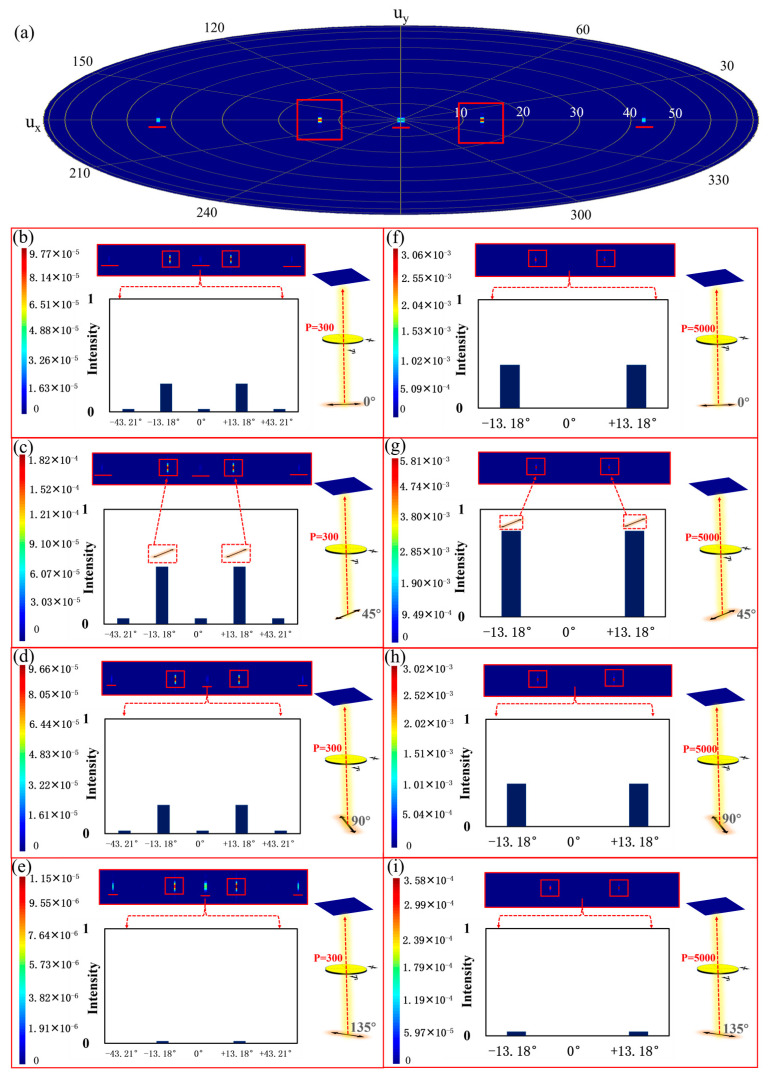
Approximate far-field projection (with a period of 300) of the NPBS for incident light sources with polarization angles of 135° (**a**). Simulated far-field transmission intensity patterns (with a period of 300) of the NPBS for incident light sources with polarization angles of 0° (**b**), 45° (**c**), 90° (**d**), and 135° (**e**). Simulated far-field transmission intensity patterns (with a period of 5000) of the NPBS for incident light sources with polarization angles of 0° (**f**), 45° (**g**), 90° (**h**), and 135° (**i**).

**Figure 7 nanomaterials-13-02592-f007:**
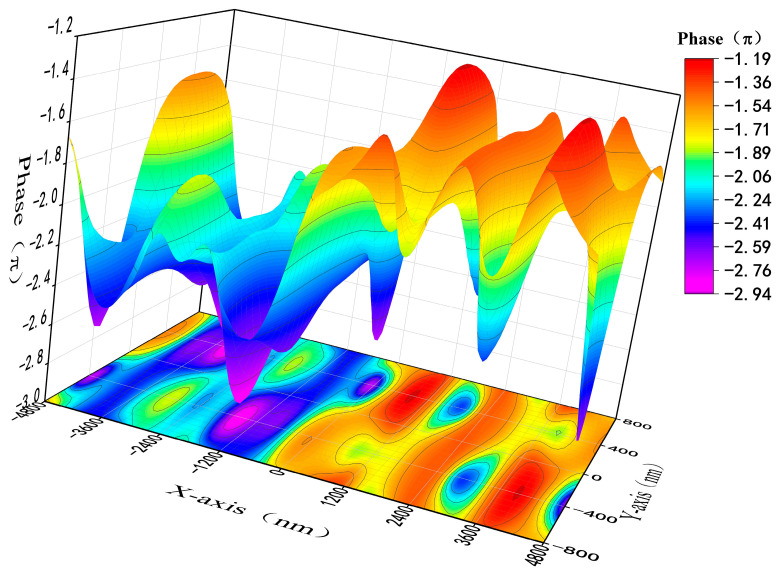
Phase distribution diagram of the left and right planes of the NPBS supercell.

**Table 1 nanomaterials-13-02592-t001:** Transmittance and ER for α = 45° and α = 135° at a wavelength of 2172 nm.

α (°)	Ta (%)	Ti (%)	ER (dB)
45	92.98	0.0207	36.53
135	93.02	0.0199	36.70

## Data Availability

The data supporting this study’s findings are available within the article.

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
