# Peer review of "Si-Based Polarizer and 1-Bit Phase-Controlled Non-Polarizing Beam Splitter-Based Integrated Metasurface for Extended Shortwave Infrared"

_nanomaterials, 2023, doi:10.3390/nano13182592_

Round 1

Reviewer 1 Report

The authors have proposed Si based linear polarizer and NPBS concept, by adopting their earlier work on GaAs metasurface design proposal. The work is relevant and detailed. I support the publication. 

Author Response

Dear Reviewer,

We sincerely appreciate your thorough review and valuable feedback. Your support for our work is of great significance to us and the publication of our research.

We will continue to exert our efforts to ensure that our work is further refined and enhanced. Again, we would like to express our gratitude for your suggestions and encouragement. Your insights will undoubtedly have a positive impact on our research.

Best regards,

Leidong Shi 

Reviewer 2 Report

A method to achieve linear polarization control is proposed based on Si nanocylinders and nanocubes on SiO2. A metasurface with the proposed unit cells, namely the nano-cube polarization conversion metal-atoms and nano-cylinder phase shift meta-atoms is further designed and demonstrated. The theoretical and numerical results confirm the validity of the proposed approach. The impact of the number of supercells in the metasurface on its performance is also discussed. Owing to the mature fabrication process of Si-based materials, the device could find applications in optics.

Overall, the manuscript is well organized and easy to follow. Adequate details are given for the theoretical calculations and numerical simulations. The results are also verified and their relevance to real-world applications is also discussed. I only have a few questions for the authors to address.

(1) The authors mentioned a prospective application of optical digital encoding. Please comment on how to possibly incorporate dynamic tuning in the proposed metasurface for digital implementation.

(2) Achieving phase control with the consideration of polarization is an active research topic, and relevant publications in this direction need to be cited, e.g., https://doi.org/10.1103/PhysRevLett.118.113901.

Minor edits may be needed to further improve the presentation of the manuscript.

Reviewer 3 Report

1. The Theoretical analysis of Section 2 seems known. Thus, the authors should be more clear on the novelties of their work. Have they fabricated any of their designs?

2. Have the authors used their in-house code for the FDTD method or a specific computational package. In the case of the former, please provide details and some validation results with the outcomes of an FDTD commercial software.

3. Moreover, it would be nice to see some more details regarding the numerical implementations.

4. The results section seems adequate, however, the authors should try to provide some comparative results regarding the efficiency of some already existing devices.

5. In the Numerical results Section the authors mention the Optimization aspect. However, it is not so clear how this is conducted for the specific problem. Perhaps they could give am ore elaborate description, a flow-chart or something more helpful for the reader.

6. A more detailed interpretation of the results in Fig.6 should be provided, while the quality of all Figures should be enhanced.

Although understandable, the language of the paper needs some polishing.

Round 2

Reviewer 4 Report

I support the article for its publication 

Author Response

Dear Reviewer,

Thank you very much for your support and positive evaluation. We are delighted to know that you endorse the publication of our article. Your feedback is highly valuable to us, and we will proceed with the publication process in accordance with the journal's requirements and your suggestions.

Once again, thank you for your support and assessment.

Sincerely,

Leidong Shi